# Restoring Balance: The Role of Omega-3 Polyunsaturated Fatty Acids on the Gut–Brain Axis and Other Interconnected Biological Pathways to Improve Depression

**DOI:** 10.3390/nu17213426

**Published:** 2025-10-31

**Authors:** Floriana De Cillis, Veronica Begni, Patrizia Genini, Daniele Leo, Marco Andrea Riva, Annamaria Cattaneo

**Affiliations:** 1Department of Pharmacological and Biomolecular Sciences, University of Milan, 20133 Milan, Italy; floriana.decillis@unimi.it (F.D.C.); patrizia.genini@unimi.it (P.G.); m.riva@unimi.it (M.A.R.); 2Biological Psychiatry Laboratory, IRCCS Istituto Centro San Giovanni di Dio Fatebenefratelli, 25125 Brescia, Italy; vbegni@fatebenefratelli.eu (V.B.); dleo@fatebenefratelli.eu (D.L.)

**Keywords:** major depressive disorder, omega-3 polyunsaturated fatty acids, gut–brain axis, inflammation, neuroplasticity, hypothalamic–pituitary–adrenal axis, oxidative stress

## Abstract

Major depressive disorder (MDD) is a complex, multifactorial condition involving dysregulation across immune, neural, and metabolic systems. Omega-3 polyunsaturated fatty acids (*n*-3 PUFAs), particularly eicosapentaenoic acid (EPA) and docosahexaenoic acid (DHA), have emerged as promising adjunctive interventions, with evidence supporting their efficacy in alleviating depressive symptoms. Here, we synthesize current knowledge on the interconnected biological pathways through which *n*-3 PUFAs may exert antidepressant effects. A growing body of evidence implicates the gut–brain axis as a central pathway through which *n*-3 PUFAs may exert antidepressant effects. By shaping microbiota composition and metabolite production, *n*-3 PUFAs influence intestinal permeability, immune activation, and vagal signaling, thereby contributing to both immunomodulatory and neurochemical effects. In combination, *n*-3 PUFAs modulate peripheral and central inflammation by promoting specialized pro-resolving mediators, downregulating pro-inflammatory cytokines, and influencing microglial activation. Parallel actions on neuronal membrane composition and lipid raft integrity affect neurotransmitter signaling, synaptic plasticity, and neurogenesis, with downstream effects on neural function. Additional pathways, including hypothalamic–pituitary–adrenal axis regulation and oxidative stress reduction, further integrate *n*-3 PUFA actions across multiple systems. Collectively, these mechanisms suggest that *n*-3 PUFAs act as network modulators, supporting recovery in depression. Translational research highlights the importance of EPA-predominant formulations, optimal dosing, and patient stratification. By framing *n*-3 PUFAs activity within a multi-level systems biology perspective, this review provides a comprehensive mechanistic understanding and underscores their potential as targeted adjunctive strategies for MDD.

## 1. Introduction

Major depressive disorder (MDD) is a leading cause of disability, yet therapeutic progress is limited for patients who fail first-line pharmacotherapy. Converging evidence across psychiatry, immunology, and metabolism research points to an impaired capacity for inflammatory resolution, with low-grade inflammation and blunted plasticity as recurrent features [1,2,3]. Increasingly, the gut–brain axis has emerged as a critical interface in this process, since dysbiosis, intestinal barrier dysfunction, and altered microbial metabolite production contribute to systemic immune activation and neuroinflammatory cascades that disrupt mood-regulating circuits [4]. In this context, interest has grown in interventions targeting these pathophysiological mechanisms. Long-chain omega-3 polyunsaturated fatty acids (*n*-3 PUFAs), notably eicosapentaenoic acid (EPA) and docosahexaenoic acid (DHA), map onto this biology, thereby linking modulation of the gut–brain axis with immune resolution and synaptic function into a unified framework of antidepressant action. Indeed, EPA and DHA are integral constituents of phospholipid membranes in all cells, where they influence membrane fluidity, receptor signaling, and ion channel dynamics, underlining their indispensable role in maintaining cell integrity and communication.

Individuals with MDD often show reduced cellular and plasma levels of EPA and DHA [5], suggesting that inadequate incorporation of these fatty acids into neuronal and peripheral membranes may contribute to mood dysregulation. Accordingly, different studies indicate small-to-moderate antidepressant effects of *n*-3 PUFAs, with EPA-rich formulations proving more effective than DHA, particularly when used as an adjunct to antidepressants. A meta-analysis and meta-regression of 13 randomized controlled trials (RCTs) showed greater benefit at higher EPA doses and in trials permitting concomitant antidepressant use [6]. Earlier work targeting MDD similarly supported EPA over DHA as the active component [7], and a large meta-analysis confirmed an overall benefit of *n*-3 PUFAs [8]. In line, practice guidelines from the International Society for Nutritional Psychiatry Research now recommend adjunctive *n*-3 PUFAs in MDD [9]. Interestingly, *n*-3 PUFAs were also found to be able to prevent recurrence in late life depression, suggesting a preventive effect in currently euthymic MDD patients [10]. Therefore, it may be possible that low baseline *n*-3 PUFAs may be a prerequisite for observing measurable effects of supplementation in clinical trials, whereas subjects with adequate baseline levels might not display additional benefit.

Nevertheless, *n*-3 PUFAs remain under investigation as potential interventions for depression, with persistent heterogeneity across studies. Such variability is likely attributable to the diverse biological pathways engaged by these compounds. Accordingly, in this review, we integrate evidence into a mechanistic framework for how *n*-3 PUFAs may alleviate depressive symptoms. Specifically, we synthesize findings that show how *n*-3 PUFAs (i) restore gut–barrier integrity and microbiome-to-brain signaling, (ii) induce anti-inflammatory effects, (iii) support the structural and functional plasticity of the brain, (iv) modulate the hypothalamic–pituitary–adrenal (HPA) axis, and (v) recover oxidative damage.

## 2. Regulation of Microbiota–Gut–Brain Axis by Omega-3 Polyunsaturated Fatty Acids

*n*-3 PUFAs exert important effects on peripheral interfaces, particularly the intestinal barrier and gut microbiota, which regulate immune tone and generate metabolic signals that feedback to the brain. In line, an increasing number of preclinical and clinical studies indicate that the gut microbiota may act as a key mediator of the antidepressant effects of *n*-3 PUFAs [11].

The microbiota–gut–brain axis refers to the bidirectional communication network linking the intestinal microbial ecosystem with neural, endocrine, and immune pathways that regulate brain function. Consequently, this axis has emerged as a critical interface between diet, systemic physiology, and mental health, positioning *n*-3 PUFAs as modulators of microbial composition and downstream host responses [12].

Clinically, patients with depression display a markedly altered microbial profile compared with healthy controls, characterised by reduced diversity and richness [13]. This dysbiosis is typified by an increased abundance of pro-inflammatory taxa such as *Proteobacteria*, normally rare in homeostatic conditions [14], and opportunistic species within *Clostridium* and *Desulfovibrio*, concomitant with a depletion of beneficial commensals including *Lactobacillus*, *Bifidobacterium*, *Faecalibacterium prausnitzii*, and *Akkermansia muciniphila* [15,16]. Parallel findings in germ-free and fecal transplant studies confirm a causal contribution of depressive microbiota to behavioural phenotypes in rodents [17]. Such taxonomic shifts have functional consequences, including altered production of key microbial metabolites such as short chain fatty acids (SCFAs) and impaired epithelial barrier integrity. In turn, increased intestinal permeability (“leaky gut”) facilitates translocation of lipopolysaccharide (LPS), which activates Toll-like receptor 4 (TLR4) signalling, triggers pro-inflammatory cytokines, and initiates a neuroinflammatory cascade that disrupts neuronal circuits relevant to mood regulation [18,19]. These perturbations correlate with depressive symptoms in both animal models and human cohorts [20].

EPA and DHA exert direct effects on intestinal epithelial cells and the local mucosal environment. Their incorporation into epithelial membranes alters lipid raft fluidity [11,21]. In addition, *n*-3 PUFAs preserve the mucosal barrier, enhance endothelial tight junctions, increase submucosal collagen synthesis and modulate epithelial membrane architecture [22,23], strengthening the intestinal barrier and limiting peripheral immune activation.

Beyond these actions, accumulating studies suggest that *n*-3 PUFAs supplementation can beneficially reshape microbial communities, increasing diversity and fostering anti-inflammatory taxa [24]. Specifically, EPA and DHA increase the abundance of *Bifidobacterium* and *Lactobacillus*, known to dampen intestinal inflammation, thereby reducing tumor necrosis factor-α (TNF-α) and pro-inflammatory interleukin expression [25,26]. In parallel, *n*-3 PUFAs reduce pro-inflammatory *Enterobacteriaceae* and opportunistic *Clostridia*, as demonstrated in animal models and human dietary intervention studies [24,27].

*n*-3 PUFAs also promote the expansion of SCFA-producing genera such as *Blautia*, *Bacteroides*, *Roseburia*, and *Coprococcus* [28,29]. SCFAs, particularly butyrate, acetate, and propionate, support regulatory T cell function and sustain an anti-inflammatory intestinal environment [30,31]. Moreover, SCFA reinforce epithelial tight junctions (occludin, claudin-1, ZO-1), thereby reducing paracellular permeability [32,33,34,35]. Importantly, SCFAs exert also direct neuroactive effects as they cross the blood–brain barrier, promoting microglial polarization toward an anti-inflammatory phenotype [36,37], and stimulating neurotrophic factors such as Brain-Derived Neurotrophic Factor (BDNF) and Glial cell line-derived neurotrophic factor (GDNF), which sustain synaptic plasticity [38].

In addition, *n*-3 PUFAs supplementation also impacts microbial and host tryptophan metabolism, central components of the gut–brain axis, serving as a biochemical interface between the microbiome, immune system, and neural circuits. Tryptophan is the essential amino acid precursor of serotonin, but under conditions of immune activation it is preferentially degraded along the kynurenine pathway through the induction of indoleamine 2,3-dioxygenase (IDO). This diversion lowers the substrate available for serotonin synthesis and simultaneously generates neuroactive kynurenine metabolites. Clinical and preclinical studies indicate that *n*-3 PUFAs attenuate this inflammatory redirection of tryptophan metabolism. *n*-3 PUFAs suppress IDO activity, reduce the kynurenine/tryptophan ratio, and promote a shift toward serotonin biosynthesis [39,40]. By rebalancing the competition between these two pathways, *n*-3 PUFAs help restore serotonergic tone while limiting the accumulation of neurotoxic kynurenine derivatives, thereby addressing a well-characterized biochemical disturbance in MDD.

Coupled with these, *n*-3 PUFAs can modulate the vagus nerve, the principal parasympathetic pathway linking the gut and brain. By enhancing vagal afferent activity, *n*-3 PUFAs increase the transmission of anti-inflammatory and homeostatic signals from the periphery to central regulatory circuits. Preclinical and emerging clinical evidence indicates that this mechanism contributes to antidepressant effects, positioning the vagus nerve as a key neuroimmune conduit through which dietary *n*-3 PUFAs influence emotional regulation [41,42].

Animal models of depression frequently rely on exposure to various forms of stress at different temporal windows (perinatal, adolescence or adulthood) to induce behavioral and biological alterations resembling those observed in human MDD. Indeed, exposure to stress is considered a central precipitating and perpetuating factor in MDD. Accordingly, preclinical models of MDD provide important insights into the interaction between stress, gut microbiota, and *n*-3 PUFAs. For instance, Pusceddu et al. showed that long-term *n*-3 PUFAs administration restored gut microbiota composition in maternal-separated rats, normalizing the disrupted *Firmicutes/Bacteroidetes* ratio [43]. Similarly, Davis et al. reported that social isolation in adult mice altered gut microbiota, changes that were mitigated by dietary DHA. Notably, they observed sex-specific responses, with males showing greater sensitivity, including reduced SCFA-producing *Allobaculum* and increased tryptophan-metabolizing *Ruminococcus* linked to depressive-like behaviour [44]. In a model of chronic stress exposure during adulthood, *n*-3 PUFAs increased sucrose preference and duration of climbing and decreased the duration of immobility tested in the forced swim test in stress-exposed rats, likely improving depressive-like phenotype. In association, chronic stress decreased the abundance of *Ruminococcaceae*, which was instead increased by *n*-3 PUFAs [45].

Together, these findings outline that *n*-3 PUFAs remodel the gut microbiota, reinforce barrier integrity, and recalibrate host immune–neuroendocrine crosstalk. This positions *n*-3 PUFAs as promising adjunctive strategies for MDD, but robust confirmation of the microbiota’s mediating role requires well-powered randomised controlled trials incorporating stratified populations and multi-omics analyses [46,47].

## 3. Anti-Inflammatory Effects of Omega-3 Polyunsaturated Fatty Acids

Notably, many of these microbiota-driven effects converge on the regulation of inflammation, a hallmark alteration in MDD. Indeed, increased gut permeability and dysbiotic microbial shifts facilitate immune activation, fueling a pro-inflammatory milieu that propagates through the gut–brain axis. MDD has indeed been associated with a state of chronic low-grade inflammation, characterized by elevated levels of *C*-reactive protein (CRP), interleukin-6 (IL-6), TNF-α, and other pro-inflammatory mediators [48]. Different studies consistently show higher concentrations of these cytokines in patients with MDD compared to healthy controls, and their levels often correlate with illness severity and treatment resistance [48,49].

Beyond gut-mediated effects, converging human studies indicate that *n*-3 PUFAs further exert antidepressant actions largely through immunomodulation and the active resolution of inflammation. In this sense, the anti-inflammatory properties of *n*-3 PUFAs represent not a separate pathway, but rather a direct extension of their gut-mediated actions.

In a dose-finding RCT enriched for high-inflammation MDD, EPA improved depressive symptoms and modulated inflammatory biomarkers in a dose-responsive manner [50]. Notably, a priori inflammation-stratified designs and secondary analyses suggest that baseline CRP and related markers moderate antidepressant response to *n*-3 PUFAs, consistent with a “inflamed depression” subtype more likely to benefit [51].

EPA and DHA serve as substrates for specialized pro-resolving mediators (SPMs), including resolvins, protectins, and maresins, which actively terminate inflammation. In patients with MDD, EPA supplementation increases circulating SPMs precursors and products (e.g., 18-HEPE, RvE2, RvE3) in a dose- and time-dependent fashion, linking dietary intake to pro-resolving biochemistry in vivo [52]. Observational studies in adolescents further implicate the resolution pathway. Lower levels of maresin-1 are associated with greater symptom severity and normalizes with treatment, while resolvin D1 also varies with disease, supporting an imbalance between inflammation and its resolution in MDD [53,54]. These effects extend to the cellular level, where SPMs reduce NF-κB activation, downregulate pro-inflammatory cytokines (TNF-α, IL-1β, IL-6), and limit NLRP3 inflammasome activation [55].

Taken together, these findings suggest that *n*-3 PUFAs do not merely blunt inflammation, promoting microbial homeostasis, but at the same time actively restore its resolution, thereby targeting two interdependent processes at the core of MDD pathophysiology.

## 4. Role of Omega-3 Polyunsaturated Fatty Acids in Brain Plasticity and Neuronal Functioning

These effects of *n*-3 PUFAs to recalibrate gut–brain axis and consequently inflammation represent only one dimension of their action. Equally important is their role in supporting the structural and functional plasticity of the brain.

Indeed, DHA is a fundamental constituent of neuronal membranes, where it confers stability and fluidity, optimize receptor and ion channel function, and facilitate efficient neurotransmission and intracellular signalling [56]. By supporting membrane integrity and signalling efficiency, these properties underpin neuroplasticity and neurogenesis, key processes for mood regulation and cognitive function throughout life. Accordingly, low levels of DHA and EPA, frequently observed in MDD, may compromise these processes.

Moreover, recent evidence highlights the role of DHA’s bioactive metabolite synaptamide (*N*-docosahexaenoylethanolamine) in promoting neuronal differentiation, neurite outgrowth, synaptogenesis, and neurogenesis through cAMP/PKA-dependent signaling cascades leading to increased expression of neurotrophic genes and structural plasticity–related proteins [57].

Accordingly, in fat-1 transgenic mice, genetically modified to endogenously convert *n*-6 PUFAs into *n*-3s, thereby maintaining elevated DHA levels without dietary supplementation, high brain DHA levels promoted neural progenitor proliferation and differentiation, increased dendritic spine density, and improved hippocampus-dependent spatial memory [58]. In aging rodents, dietary EPA and DHA preserved neurogenesis in the dentate gyrus, reduced apoptosis and gliosis, and maintained hippocampal-dependent memory performance [59]. Similarly, in a preclinical model of accelerated aging, *n*-3 PUFAs supplementation, either alone or in combination with polyphenol- and carotenoid-rich extracts characterized by potent antioxidant and anti-inflammatory activities, prevented hippocampal-dependent memory decline, restored hippocampal expression of doublecortin, and increased cortical levels of the EPA-derived mediator 18-HEPE [60]. In a model of traumatic brain injury, the addition of *n*-3 PUFAs could normalize the levels of BDNF, synapsin I, and cAMP responsive element-binding protein (CREB), thus restoring the survival of neuronal cells [61]. In inflammatory contexts, EPA supplementation increased BDNF–TrkB signaling and rescued impaired hippocampal neurogenesis [62], highlighting the importance of adequate *n*-3 PUFAs availability for neuronal plasticity. In line, in female mice, diets characterized by high levels of *n*-6 PUFAs and insufficient *n*-3 PUFAs reduced neural progenitor proliferation and the survival of newborn neurons in the dentate gyrus, while simultaneously increasing astrocytic and microglial reactivity. These findings emphasize not only the importance of adequate *n*-3 PUFAs supply but also the detrimental effects of excess dietary *n*-6 PUFAs, which may produce vulnerabilities [63].

Neuroplasticity is particularly sensitive during development. Accordingly, in the perinatal period, adequate maternal intake of DHA and EPA supports neuronal proliferation, migration, and synaptogenesis. Experimental studies show that maternal *n*-3 PUFAs deficiency impairs hippocampal neurogenesis in pups, alters gene expression controlling neuronal differentiation, and predisposes offspring to long-term cognitive and emotional vulnerability [64]. Beyond deficiency, maternal diets disproportionately enriched in linoleic acid (*n*-6 PUFA) and deficient in α-linolenic acid (*n*-3 PUFA) reprogram offspring neurodevelopment. Eventually, adult offspring displayed structural alterations in the amygdala, suggesting that excessive *n*-6 PUFAs and insufficient *n*-3 PUFAs during gestation primes limbic circuits [65]. Adolescence represents a second sensitive period, as stress during this phase reduces hippocampal neurogenesis, dendritic complexity, and long-term potentiation, often leading to impairments in memory and affect. *n*-3 PUFAs enriched diets mitigate these neurogenic and behavioural deficits [66] and preserve white-matter microstructure, as shown in rats where DHA deficiency during preadolescence disrupted forebrain tract integrity while DHA supplementation preserved connectivity [67].

In vitro studies add mechanistic detail. EPA and DHA have been shown to promote neural stem cell differentiation through upregulation of neuronal markers such as NeuroD, Tuj-1, and MAP2, while suppressing pro-apoptotic signaling [68]. In human hippocampal progenitor cells, EPA and DHA counteracted cortisol-induced suppression of proliferation and differentiation, although their timing of action differed. EPA was effective during progenitor proliferation, whereas DHA required continuous exposure [69]. Pro-inflammatory cytokines impair neurogenesis and promote apoptosis in hippocampal cultures, effects attenuated by EPA/DHA-derived SPMs [69,70]. Indeed, SPMs, generated through LOX, COX, and CYP450 pathways, exert potent anti-inflammatory and pro-neurogenic actions [71].

Building on these general findings, a growing body of research has specifically examined neuroplasticity in the context of depression, where preclinical models of stress and clinical trials in patients provide converging evidence that *n*-3 PUFAs can modulate neuronal signaling pathways relevant for mood regulation.

In an animal model of depression, Larrieu et al. evaluated whether *n*-3 PUFAs supplementation could attenuate the effects of chronic social defeat stress. The dietary intervention failed to reduce social avoidance while attenuating social defeat-induced anxiety-like behavior. Coupled with this, the complexity of neuronal dendrites in the PFC was improved by *n*-3 PUFAs supplementation [72]. Similarly, separation from the pups, mimicking postpartum depression, induced depressive behaviors. *n*-3 PUFAs supplementation reversed such phenotype and increased hippocampal expression of CREB and BDNF [73].

Clinically, in an RCT, children and adolescents with MDD were administered with a 12-week *n*-3 PUFAs supplementation. Baseline data indicated that a higher proportion of *n*-6 PUFAs was associated with lower BDNF concentrations while the supplementation with *n*-3 PUFAs increased peripheral BDNF levels [74].

## 5. Modulation of Cortisol and Stress Reactivity by Omega-3 Polyunsaturated Fatty Acids

Among the biological pathways through which *n*-3 PUFAs may influence depression, regulation of the HPA axis has attracted growing interest. Indeed, this system, which integrates stress responses with immune circuits, provides a complementary mechanism to the anti-inflammatory actions of *n*-3 PUFAs.

Evidence from animal models demonstrates that diets enriched with DHA or EPA blunt increases in corticosterone induced by stress exposure during adolescence [75]. Similarly, in a chronic stress paradigm during adulthood, EPA exerts robust effects on HPA axis regulation and stress responsivity. It reverses behavioural deficits and normalizes corticosterone levels, while concurrently modulating hippocampal inflammation, indicating that improvements in stress-axis function are closely linked to immune regulation [76]. Conversely, Kikuchi et al. (2024) showed that maternal diets during gestation and early lactation characterized by high linoleic acid (*n*-6 PUFA) and low α-linolenic acid (*n*-3 PUFA) induced persistent alterations in adult offspring, including heightened stress vulnerability and dysregulated corticosterone responses, highlighting the long-term impact of *n*-3 PUFAs on HPA axis modulation [65]. Likewise, *n*-3 PUFA deficiency, experienced either until adolescence or adulthood, produces a depressive-like phenotype characterized by HPA axis hyperactivation and elevated pro-inflammatory cytokines at both timepoints, underscoring the critical role of *n*-3 PUFA in maintaining proper stress responses and mood stability [77]. Four weeks of flaxseed oil supplementation, rich in α-linolenic acid, attenuated cortisol responses to social challenges in middle-aged guinea pigs. Specifically, the changes in HPA axis reactivity were associated with sex-specific alterations in social hierarchy, indicating that *n*-3 PUFAs primarily modulate stress physiology, with downstream effects on social behaviours [78].

Clinical evidence mirrors these preclinical observations and highlights how *n*-3 PUFAs supplementation can recalibrate stress-related neuroendocrine and immune responses. In a randomized double-blind trial of adolescents with MDD, 12 weeks of EPA and DHA supplementation (2.4 g/day) significantly reduced morning salivary cortisol compared with baseline, while an *n*-6 comparator had no effect. Notably, higher baseline cortisol was associated with greater symptom severity, suggesting that *n*-3 PUFAs may be particularly beneficial in patients characterized by HPA axis overactivity [79].

In a study of middle-aged adults exposed to a standardized laboratory stressor, four months of high-dose *n*-3 PUFAs supplementation (2.5 g/day EPA+DHA) markedly attenuated cortisol responses. This primary modulation of HPA axis activity was accompanied by blunted IL-6 responses and maintenance of IL-10 levels in the post-stress period [80]. Similarly, in a trial involving employees with clinical burnout, eight weeks of *n*-3 PUFAs treatment (1.2 g/day EPA+DHA) not only reduced emotional exhaustion and depersonalization but also enhanced personal accomplishment compared with placebo. Biologically, these improvements coincided with a significant reduction in the cortisol awakening response, aligning symptomatic relief with normalization of HPA axis activity [81].

Collectively, these findings show that *n*-3 PUFAs dampen stress-related biological activation and enhance resilience, modulating neuroendocrine–immune interactions, thus providing a plausible pathway through which supplementation may contribute to mood improvement.

Notably, the changes observed in preclinical studies on HPA functioning were often accompanied by neuronal adaptations, including restoration of astrocytic BDNF–TrkB signalling and engagement of related molecular pathways, indicating that HPA axis normalization occurs alongside neural regulation [76]. Conversely, *n*-3 PUFA deficiency induces elevated pro-inflammatory cytokines associated with disrupted neuronal signalling, further underscoring the integrated role of *n*-3 PUFA in coordinating stress-axis and neural function for mood stability [77].

## 6. Modulation of Oxidative Stress by *n*-3 Polyunsaturated Fatty Acids in the Pathophysiology of Depression

In this context, oxidative stress has emerged as a central hub, linking immune activation and impaired neuroplasticity, and provides a unifying framework through which metabolic imbalance contributes to depressive pathology. Redox disturbances are indeed tightly connected with other biological processes that characterise depression. Neuroinflammation, impaired neurotransmission and reduced neurogenesis can all be exacerbated by oxidative stress, positioning it as a critical upstream driver of dysfunction.

Oxidative stress develops when reactive oxygen and nitrogen species (ROS/RNS), normally essential for signalling and host defence, accumulate more than antioxidant defences. This imbalance transforms physiological messengers into cytotoxic agents, triggering lipid peroxidation, protein carbonylation, and oxidative DNA damage that ultimately drive neuronal apoptosis [82]. Such mechanisms are not merely theoretical as clinical studies consistently demonstrate that patients with depression exhibit biochemical signatures of oxidative damage together with reduced antioxidant capacity [83,84]. Consistent with this, individuals with depression often present with inadequate intake of antioxidant-rich micronutrients (vitamins A, C, E; selenium; zinc; B vitamins) [85], which may further weaken endogenous defences. At the biochemical level, this vulnerability manifests as enhanced lipid peroxidation, nitric oxide overproduction, and COX-2 hyperactivity, in parallel with diminished antioxidant enzyme activity [86]. Importantly, these molecular perturbations are measurable. Among the most reproducible markers are 8-hydroxy-2′-deoxyguanosine (8-OHdG), reflecting DNA oxidation, and F2-isoprostanes, stable products of lipid peroxidation. Both have been found consistently elevated in depressed patients compared with healthy controls [87,88,89], underscoring oxidative stress as a quantifiable correlate of depressive pathology.

*n*-3 PUFAs were found able to reduce lipid peroxidation in both plasma and cell membranes while enhancing antioxidant capacity in the membranes in a RCT of adolescent depression. Importantly, such modulations were associated with improvements in depressive symptoms and cognitive function [90]. Notably, the clinical efficacy of *n*-3 PUFAs seems to depend on the redox milieu in which they act. Elevated baseline oxidative stress and sufficient plasma EPA+DHA reserves predict stronger antidepressant responses to supplementation [90,91,92,93].

Preclinically, depressive-like behaviour induced by maternal deprivation were prevented by *n*-3 PUFAs supplementation, which also exerted antioxidant effects in the prefrontal cortex, hippocampus and amygdala [94]. Likewise, chronic mild stress-induced anhedonic phenotype and lipid peroxidation were completely inhibited by *n*-3 PUFAs in adult mice [95].

Similarly, but in different contexts, it has been shown that *n*-3 PUFAs can exert antioxidative effects. In a model of traumatic brain injury, *n*-3 PUFAs inclusion reduced oxidative damage [61], while in an accelerated-aging model, supplementation increased the expression of antioxidative markers, such as catalase, in the hippocampus [60].

Within this interconnected network, *n*-3 PUFAs act as modulators capable of reshaping the oxidative–inflammatory interface. Indeed, DHA suppresses microglial oxidative and pro-inflammatory activity via HO-1 upregulation [96], while both EPA and DHA activate the Nrf2 transcription factor, inducing broad cytoprotective gene programs [69,97]. In addition, ROS can potentiate HPA-axis hyperactivation [98], while glucocorticoids in turn amplify ROS production [99,100], creating a vicious cycle confirmed experimentally following corticosterone administration [101].

However, it is important to recognize that, due to their high degree of unsaturation, *n*-3 PUFAs may also promote peroxidation at high concentrations. In turn, this characteristic underlies a bimodal effect on oxidative balance, where moderate doses of *n*-3 PUFAs enhance antioxidant defences, while excessive intakes increase lipid peroxidation. Indeed, high intakes of DHA increased markers of oxidative stress such as urinary isoprostane that was instead lowered by low consumption of DHA [102].

Taken together, these findings establish oxidative stress as a core node in the pathophysiology of depression, intricately connected with neuroinflammation, HPA-axis dysregulation, and maladaptive plasticity [103]. By virtue of their pleiotropic antioxidant and anti-inflammatory properties, omega-3 PUFAs are uniquely positioned to target this intersection.

## 7. Conclusions

*n*-3 PUFAs influence depression by modulating multiple, interconnected pathways, including gut microbiota composition, neuroplasticity, HPA-axis regulation and redox balance (Figure 1). Rather than acting on a single target, they help restore overall homeostasis across endocrine, immune, metabolic, and neural networks. This integrative view emphasizes that EPA and DHA are essential biological molecules whose adequate availability is critical for maintaining neuronal, immune, and metabolic health. MDD may, in part, reflect a state of deficiency of these fatty acids, explaining why supplementation is most effective in individuals with low baseline levels.

However, translating these mechanistic insights into consistent clinical benefits has proven to be challenging. Treatment response appears to depend on a combination of baseline factors, including inflammatory burden, metabolic status, habitual diet, erythrocyte PUFAs composition, genetic variants in fatty acid metabolism, microbiome profiles, sex, life stage, and concurrent antidepressant therapy.

Among these, the broader dietary context may play a critical role. Western diets are typically characterized by a high intake of *n*-6 PUFAs and a low intake of *n*-3 PUFAs, which promotes intestinal permeability, endotoxemia, and systemic inflammation. In animal models, such conditions amplify circulating LPS and its binding protein, effects that can be reversed by either antibiotics or *n*-3 supplementation, partly through induction of intestinal alkaline phosphatase, an enzyme that detoxifies LPS [104,105]. Excess *n*-6 PUFAs also compete with *n*-3s for shared metabolic enzymes, limiting their conversion into SPMs [106,107]. These findings suggest that *n*-3 PUFAs efficacy may depend less on absolute dose than on the surrounding nutritional and metabolic landscape, a factor often overlooked in clinical trials. Another factor often overlooked is the bioavailability of *n*-3 PUFAs that can be influenced by several exogenous (i.e., chemical form of *n*-3 PUFAs, food matrix and processing methods) and endogenous (i.e., health of the gastrointestinal tract) factors [108].

Additional variability in outcomes likely reflects differences in baseline redox–inflammatory status and gut microbiome composition, highlighting the potential of biomarker-guided stratified approaches. However, few clinical trials have prospectively stratified participants by inflammatory markers, genetic polymorphisms affecting fatty acid metabolism or microbiome profiles.

Additionally, most studies are short-term (≤12 months), leaving important questions about long-term efficacy, optimal treatment duration, maintenance dosing, withdrawal effects, and interactions with conventional antidepressants unanswered. Safety considerations are also critical, particularly for high-dose supplementation (>3 g/day) in older adults or individuals with comorbidities.

Although evidence favours EPA over DHA for antidepressant effects, the optimal formulation, including the EPA:DHA ratio, and the role of other *n*-3 PUFAs (e.g., docosapentaenoic acid) remains unclear. Current dosing recommendations are largely empirical, as dose–response studies powered to detect efficacy across patient subgroups are lacking.

Overall, *n*-3 PUFAs represent a biologically grounded adjunct for depression, but their clinical utility likely depends on patient-specific factors and the nutritional-metabolic context. Future research should integrate multi-omics tools, identify validated biomarkers, and design biomarker-stratified trials to determine who is most likely to benefit, under which conditions, and for how long.

Omega-3 fatty acids engage multiple signalling cascades affecting neuroimmune, neuroendocrine, and oxidative stress pathways. By modulating cytokine networks, cortisol responsiveness, ROS dynamics, and gut–brain interactions, they sustain neuronal function and synaptic plasticity under physiological and stress conditions.

## Figures and Tables

**Figure 1 nutrients-17-03426-f001:**
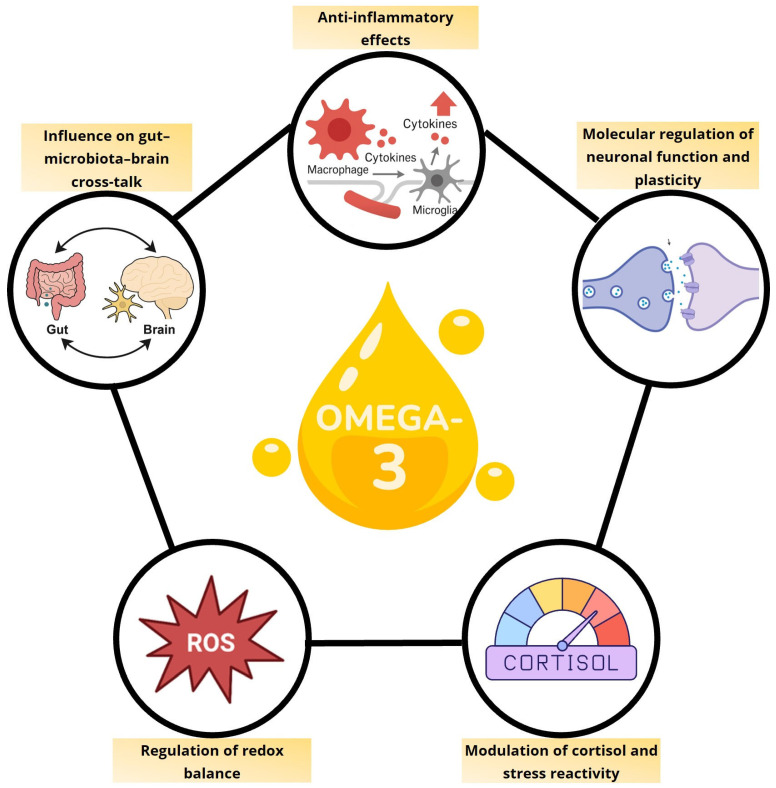
Integrated pathways mediating the neurobiological effects of omega-3 fatty acids.

## Data Availability

No new data were created or analyzed in this study. Data sharing is not applicable to this article.

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
