# Peer review of "Restoring Balance: The Role of Omega-3 Polyunsaturated Fatty Acids on the Gut–Brain Axis and Other Interconnected Biological Pathways to Improve Depression"

_nutrients, 2025, doi:10.3390/nu17213426_

Round 1
Reviewer 1 Report
Comments and Suggestions for Authors
De Cillis et al provide a narrative review on the role of omega-3 fatty acids in major depression, highlighting the roles of the gut, anti-inflammatory actions, brain structure and function, the HPA-axis, and antioxidative properties.
In introduction, the scope of the review is defined. Then, the topics are discussed in a scholarly fashion, leaving little room for improvement, except: in line 275, 391, 394, and others, the authors mention “an n-6:n-3 ratio”. Since it has never been defined, what individual fatty acids are mentioned by this ratio, and for other reasons (PMID 18039412), a ratio n-6:n-3 should not be mentioned in a scientific publication.
However, the review was written based on the concept that EPA&DHA rather are pharmacologic compounds than substances intricate to human life. EPA&DHA are essential parts of the cell membrane, and no human being has been found without minimal cellular levels of EPA&DHA. Cillis et al only briefly allude to EPA&DHA in cell membranes (lines 104 – 109), but otherwise ignore these aspects. Levels of EPA&DHA in persons with major depression are low (PMID 33806218). Major depression has a diverse pathophysiology, as the authors correctly state in the first para of introduction, with low cellular levels of EPA&DHA being a common denominator. Low levels of EPA&DHA are a prerequisite for effects of EPA&DHA to become detectable in an intervention trial, and in individuals with levels in the target range, no effect will be observed. Moreover, EPA&DHA have a complex bioavailability (PMID 39736417). These facts bear on study design, interpretation of results, and use of EPA&DHA in clinical practice. This reviewer suggests re-writing the review with these remarks in mind.
Author Response
Reviewer 1
De Cillis et al provide a narrative review on the role of omega-3 fatty acids in major depression, highlighting the roles of the gut, anti-inflammatory actions, brain structure and function, the HPA-axis, and antioxidative properties.
In introduction, the scope of the review is defined. Then, the topics are discussed in a scholarly fashion, leaving little room for improvement, except: in line 275, 391, 394, and others, the authors mention “an n-6:n-3 ratio”. Since it has never been defined, what individual fatty acids are mentioned by this ratio, and for other reasons (PMID 18039412), a ratio n-6:n-3 should not be mentioned in a scientific publication.
We thank the reviewer for the positive feedback and valuable comment. As indicated by the reviewer and since the ratio may not accurately represent the complexity of fatty acid metabolism and interactions, we have removed all mentions of the “n-6:n-3 ratio” throughout the manuscript.
However, the review was written based on the concept that EPA&DHA rather are pharmacologic compounds than substances intricate to human life. EPA&DHA are essential parts of the cell membrane, and no human being has been found without minimal cellular levels of EPA&DHA. Cillis et al only briefly allude to EPA&DHA in cell membranes (lines 104 – 109), but otherwise ignore these aspects. Levels of EPA&DHA in persons with major depression are low (PMID 33806218). Major depression has a diverse pathophysiology, as the authors correctly state in the first para of introduction, with low cellular levels of EPA&DHA being a common denominator. Low levels of EPA&DHA are a prerequisite for effects of EPA&DHA to become detectable in an intervention trial, and in individuals with levels in the target range, no effect will be observed. Moreover, EPA&DHA have a complex bioavailability (PMID 39736417). These facts bear on study design, interpretation of results, and use of EPA&DHA in clinical practice. This reviewer suggests re-writing the review with these remarks in mind.
We thank the reviewer for this insightful and constructive comment. We fully agree that EPA and DHA should not be viewed merely as pharmacologic compounds, but rather as essential structural and functional components of human cell membranes. In line with this suggestion, we have revised sections 1 (lines 57-63, 73-76), 4 (lines 222-223) and 7 (lines 408-412, 427-430).
Reviewer 2 Report
Comments and Suggestions for Authors
See comments above and in the attached file.
